# The Development of Classical Swine Fever Marker Vaccines in Recent Years

**DOI:** 10.3390/vaccines10040603

**Published:** 2022-04-13

**Authors:** Fangfang Li, Bingke Li, Xinni Niu, Wenxian Chen, Yuwan Li, Keke Wu, Xiaowen Li, Hongxing Ding, Mingqiu Zhao, Jinding Chen, Lin Yi

**Affiliations:** 1College of Veterinary Medicine, South China Agricultural University, No. 483 Wushan Road, Tianhe District, Guangzhou 510642, China; fangfangli@stu.scau.edu.cn (F.L.); 20193073038@stu.scau.edu.cn (B.L.); iniui648@stu.scau.edu.cn (X.N.); chwenxian0912@163.com (W.C.); 20191028012@stu.scau.edu.cn (Y.L.); wukeke@stu.scau.edu.cn (K.W.); xiaowenlee@stu.scau.edu.cn (X.L.); dinghx@scau.edu.cn (H.D.); zmingqiu@scau.edu.cn (M.Z.); 2Guangdong Laboratory for Lingnan Modern Agriculture, College of Veterinary Medicine, South China Agricultural University, Guangzhou 510642, China; 3Key Laboratory of Zoonosis Prevention and Control of Guangdong Province, Guangzhou 510642, China

**Keywords:** CSF, DIVA, vaccine design

## Abstract

Classical swine fever (CSF) is a severe disease that has caused serious economic losses for the global pig industry and is widely prevalent worldwide. In recent decades, CSF has been effectively controlled through compulsory vaccination with a live CSF vaccine (C strain). It has been successfully eradicated in some countries or regions. However, the re-emergence of CSF in Japan and Romania, where it had been eradicated, has brought increased attention to the disease. Because the traditional C-strain vaccine cannot distinguish between vaccinated and infected animals (DIVA), this makes it difficult to fight CSF. The emergence of marker vaccines is considered to be an effective strategy for the decontamination of CSF. This paper summarizes the progress of the new CSF marker vaccine and provides a detailed overview of the vaccine design ideas and immunization effects. It also provides a methodology for the development of a new generation of vaccines for CSF and vaccine development for other significant epidemics.

## 1. Introduction

Classical swine fever (CSF) is an acute, febrile, and highly fatal infectious disease for pigs caused by the classical swine fever virus (CSFV) [1]. It is listed by the World Organization for Animal Health (OIE) as an infectious animal disease that must be reported [2]. CSFV, bovine viral diarrhea mucosal disease virus (BVDV), and boundary disease virus (BDV) belong to the family Flaviridae and the plague virus [3]. CSFV is a single-stranded positive-stranded RNA virus with about 12.5 kb, which contains a complete open reading frame (ORF) and encodes a precursor polyprotein, which is processed into four structural proteins (C, E^rns^, E1, and E2) and eight non-structural proteins (N^pro^, p7, NS2, NS3, NS4A, NS4B, NS5A, and NS5B) under the action of virus-specific proteases and cytoproteases. The position of each protein in the virus genome from the N-terminal to C-terminal was N^pro^, C, E^rns^, E1, E2, p7, NS2, NS3, NS4A, NS4B, NS5A, and NS5B [4]. The E2 protein was the dominant candidate for the development of subunit vaccines because it has high immunogenicity and induces the production of high levels of neutralizing antibodies [5,6].

Currently, CSFV is mainly found in Central and South America, Europe, Asia, and parts of Africa (Figure 1). The primary control strategy for CSF in China is currently a combination of bonification and vaccination. Bonification is appropriate for regions where CSF was eradicated or had never occurred; however, for a country as large as China that raises a lot of pigs, vaccination is both more cost-effective and economical. Pig herds that are intensive farmed already have elemental immune resistance for CSF vaccination. Therefore, the immunosuppression caused by the sporadic epidemic of mild CSF and atypical CSF also brings significant challenges to the prevention and the control of the disease [7]. It is particularly noteworthy that moderately virulent or attenuated CSFV strains can cause persistent infection and recessive infection, immunosuppression, and virus-carrying syndrome in susceptible pigs, resulting in immune dysfunction and immune failure [8,9,10,11]. On the one hand, CSFV infection causes apoptosis of the body’s immune cells, thereby inhibiting dendritic cell maturation and antigen-presenting ability [12]; on the other hand, autophagy induced by CSFV infection can not only induce virus replication, but also inhibit apoptosis by down-regulating RIG-I-like receptor (RLR) signaling, resulting in persistent infection by CSFV in the host [13,14]. It has also been shown that CSFV infection suppresses the level of interferon (IFN) secretion, leading to persistent CSFV infection [15].

Since the first case of African classical swine fever (ASF) was diagnosed in Shenyang on 3 August 2018, it has rapidly swept the country, causing tens of billions of direct economic losses [16]. In contrast, the clinical symptoms and pathological changes of CSF and ASF are incredibly similar and can interfere with each other’s diagnosis. Good prevention and control of ASF facilitate CSF decontamination, and good immunization and prevention and control of CSF allow for better diagnosis, prevention, and control of ASF. The importance of biosecurity and CSF control and purification is emphasized in the context of the lack of an effective vaccine for ASF and the shift from typical to atypical (mild) CSF. Therefore, the development of a new type of CSF vaccine should have the immune effect of a C strain live-attenuated vaccine and distinguish between wild virus and vaccine virus infection. Currently, various research institutes are studying improved live virus vaccines, subunit vaccines, marker vaccines, etc. The purpose of this review is to present progress in CSF vaccine research and to suggest a possible methodology for targeting the development of vaccines against CSF and other significant epidemics.

## 2. Live-Attenuated Vaccines (LAVs)

The live-attenuated vaccine was obtained from highly virulent strains weakened by successive generations of rabbits or cells. Live-attenuated vaccines have been used for mass vaccination for more than 50 years. Currently, in everyday use are the Chinese vaccine strain (C-strain), the GPE strain from Japan, the Thiverval strain from France, and the PAV-250 strain from Mexico [7,17].

The C-strain is genetically stable, safe, and effective, and induces an immune response that is resistant to strains of different genotypes, resulting in extensive cross-reactivity [18]. The C-strain provides adequate protection for immunized pigs within four days, and the duration of immunity is long enough to protect for at least 6–11 months [19]. Some studies have shown that vaccination with the C strain stimulates the proliferation of helper T cells [18]. In addition, the C strain virus replicates very slowly in lymphoid tissues. It thus temporarily evades recognition by toll-like receptors (TLRs), avoiding overexpression of IFNs and pro-inflammatory cytokines and indirectly protecting immune cells and organs. When the virus accumulates to a certain level, the expression of TLRs is induced, the innate immune system is initiated, and the virus is cleared [20]. In addition to intramuscular injection, oral immunization can also be taken orally. In a study in Thailand, the oral form of the CSF vaccine was modified for its dependence on storage temperature by using bread as a substrate to carry the lyophilized CSF vaccine, which was better absorbed and maintained the virus titer. Seroconversion rates were 90% in piglets of different ages at 14 days after immunization and could reach 100% by 28 days [21,22]. By the end of 2020, there were 13 commercial live vaccines of classical swine fever in China, among which there are single vaccines, combined vaccines, and triple vaccines. In addition, other strains derived from the C strain are also widely used abroad.

The Thiverval strain is derived from the Alfort strain that has been continuously passed through more than 170 generations at 29–30 °C. Studies have shown that the Thiverval strain provides complete protection 5 days after inoculation, produces neutralizing antibodies 14 days after inoculation, and effectively inhibits the replication of CSFV after the attack [23]. GPE strain has 225 mutations compared with its parent ALD strain, which is also passed continuously at 30 °C [24]. PAV-250 is derived from the A-PAV-1 strain with more than 250 consecutive passages and is approved for use in Mexico [25]. 

All of the above vaccines can provide complete protection to the organism. Still, the weak vaccine does not have immunological markers to distinguish between vaccination and wild virus infection, posing a severe challenge to the eradication of CSF.

## 3. DIVA Vaccines

The C strain is the most widely used live-attenuated vaccine for controlling CSF. Still, the antibody response induced by the C strain is indistinguishable from that caused by infection with the wild virus. By serological analysis, marker vaccines (DIVA) can differentiate between wild virus-infected and vaccinated animals. Currently, the DIVA vaccine has been used to eradicate pseudorabies and avian influenza [26,27]. Thus, it is urgent to develop a DIVA vaccine that can effectively differentiate between vaccinated and infected animals and has comparable immune efficacy to the C strain. Live-attenuated vaccines, viral vector vaccines, and subunit vaccines are often used to develop marker vaccines for CSF. The following section describes these three aspects.

### 3.1. MLV Vaccine

The genetic construction of chimeric plague viruses has proven to be a promising strategy for generating highly efficient MLVs with marker properties. A new CP7_E2Alf (produced by the Zoetis company, under the name “Suvaxyn^®^CSF Marker”) marker vaccine was approved by the European Medicines Agency in 2014 [28]. CP7_E2Alf is a plague virus chimera embedded in the BVDV virus backbone, constructed by replacing the E2 gene of BVDV with the E2 gene of Alfort/187 strain CSFV. The CP7_E2alf vaccine effectively protected pigs against strong CSFV attacks, both intramuscularly and orally, provided complete protection for pregnant sows and their piglets, and was resistant to wild strains of genotypes 2.1 and 2.3 [29]. However, due to the close antigenic relationship between CSFV and BVDV, cross-reactive antibodies may lead to false-positive reactions and failure of DIVA recognition [30,31]. Additionally, its suitability as a DIVA vaccine was dependent on E^rns^ ELISA, which could only be used at the population level due to its limited specificity and sensitivity [32,33]. Researchers have carried out an extensive study to address this issue. Postel et al., replaced the E^rns^ coding sequence of the Alfort-Tübingen strain CSFV with the homologous sequence of plague virus that was more distantly related to CSFV to construct three chimeric viruses, “Ra”, “Pro”, and “RaPro”. These three chimeric viruses contain the Erns sequences of Norway rat and Pronghorn pestiviruses, respectively, or a combination of both [34]. All three chimera-immunized animals were protected against CSFV infection. Most importantly, no cross-reactivity was seen in the serological diagnosis of the immunized animals.

In 2017, South Korea approved another DIVA vaccine, named FLC-LOM-BE^rns^, a chimeric plague virus based on the live-attenuated classical swine fever vaccine LOM. The FLC-LOM-BE^rns^ was constructed by replacing the 30-end part of the capsid protein and the full-length E^rns^ glycoprotein from the KD26 strain BVDV using the LOM CSFV backbone [35]. Sows were immunized with FLC-LOM-BE^rns^ three weeks before insemination and challenged during pregnancy. The results showed that FLC-LOM-BE^rns^ could provide complete protection for pregnant sows and prevent vertical transmission. Compared with pigs inoculated with LOM, the feed intake and weight gain of pigs inoculated with FLC-LOM-BE^rns^ were normal. The average slaughtering date was eight days earlier, and the productivity was significantly increased [36]. In addition, immune and infected animals could be distinguished by detecting antibodies against CSFV and BVDV E^rns^ proteins.

Since the C strain cannot distinguish infection from serological inoculation, Han et al., established a marker C strain vaccine conforming to the DIVA principle. Three mutant strains rHCLV-E2F117A, rHCLV-E2G119A, and rHCLV-E2P122A were constructed by replacing the corresponding regions of pHCLV with three fusion genes containing single amino acid mutations 117F, 119G, or 122P on the mAb HQ06 recognition epitope [37]. Intramuscular injection of rHCLV-E2P122A could induce anti-CSFV neutralizing antibodies 28 days after inoculation, but could not induce antibodies against HQ06 recognition epitopes. Therefore, the animals immunized with the C strain could be distinguished by detecting antibodies against the mAb HQ06-recognized epitope of monoclonal antibodies.

Another interesting study developed a double antigenic marker live-attenuated CSFV strain FlagT4v by deleting a highly conserved specific epitope of CSFV and inserting a synthetic epitope Flag^®^ [38]. The marker vaccine distinguishes vaccinated animals from infected animals by the serological response to the Flag epitope and the mAbWH303 epitope. FlagT4v vaccine given intranasally at 3 dpi or intramuscularly at 2 dpi is effective against the highly virulent BICv (Brescia strain). However, the vaccine reverted to virulence during successive generations of piglets. The researchers made improvements and constructed a new virus with higher genetic stability, FlagT4Gv, by codon transformation [39]. FlagT4Gv provided adequate protection against classical swine fever virus attack at seven days post-vaccination. In another vaccine evaluation trial, FlagT4Gv vaccination produced protection against CSFV within three days, with a significant increase in IFN-α levels [40]. IFN-α provides direct protection against strong strains of CSF [41].

### 3.2. Viral Vector Vaccine

Viral vectors are effective for expressing foreign proteins [42,43,44]. By inserting exogenous protective antigen genes into the viral genome to obtain a recombinant virus, immunization of animals will induce an immune response in the organism. This retains some of the advantages of live vaccines. Some viruses have been explored as replication vectors for CSFV vaccines.

A recombinant adenovirus is a universal gene delivery and expression system [45]. Adenovirus has the advantages of a broad host range, lack of pre-existing maternal antibodies, etc., and most importantly, it does not integrate into the host genome. Studies have shown that adenovirus was a good vector for vaccine preparation [46,47]. Sun and his team members constructed recombinant human adenovirus type 5 (RADV2-E2) expressing the CSFV E2 gene [48]. After immunizing rabbits and pigs, a high level of CSFV-specific neutralizing antibody was produced. The average protective titer of the induced antibody was 1:400, far higher than the lowest protective titer of CSF (1:50). However, the vaccine induced insufficient protection and delayed antibody response. They previously reported that a Semliki Forest virus (SFV) replicon-vectored plasmid DNA vaccine (pSFV1CS-E2) encoding the *E2* gene of CSFV induced specific immunity and protection against CSFV in pigs. However, it has the disadvantage of low gene transfer efficiency [49,50]. Therefore, they developed an adenovirus /SFV replicon chimeric vector vaccine (rAdV-SFV-E2) [51]. The vaccine induced antibodies specific to CSFV and provided aseptic immunity and complete protection against lethal attack. Then, the author made a comprehensive evaluation of the vaccine. The results showed that: (1) the vaccine was safe for mice, rabbits, and pigs; (2) two immunizations with a dose as low as 6.25 × 10^5^ TCID_50_ or a single immunization with a dose of 10^7^ TCID_50_ rAdV-SFV-E2 provided complete protection against a lethal CSFV challenge; (3) maternally derived antibodies had no inhibitory effects on the efficacy of the vaccine; (4) the vaccine did not induce interfering anti-vector immunity; (5) rAdV-SFV-E2 combined with a pseudorabies vaccine did not interfere with each other [52]. These experimental data indicate that the chimeric vector vaccine rAdV-SFV-E2 is a promising CSF marker vaccine.

Various gene-deleted pseudorabies virus (PRV) recombinants have been used as vaccine vectors expressing foreign genes [53,54,55], and could be used to develop polyvalent vaccines against pseudorabies (PR) and other diseases [56]. It has been reported that gE, gI, thymidine kinase (TK), and protein kinase (PK), which were related to the virulence of pseudorabies virus, could be replaced by foreign genes without affecting the infectivity and reproduction of the virus and its immunogenicity [57,58]. In previous studies, rPRVTJ-del*gE*/*gI*/*TK* mutants were safe for susceptible animals [57]. A recombinant virus, rPRVTJ-del*gE*/*gI*/*TK*-E2, expressing CSFV E2 protein was constructed, and its safety and immunogenicity in pigs were evaluated [59]. The results showed that the pigs inoculated with 10^6^ TCID_50_ rPRVTJ-del*gE*/*gI*/*TK*-E2 produced E2-specific antibodies at the same level as the C strain group after seven days of enhanced immunization, and had complete protection against the deadly shimen strain of CSFV and variant PRVTJ strain, and no virus drop was detected. These data suggest that the recombinant virus rPRVTJ-del*gE*/*gI*/*TK*-E2 could be used to develop bivalent vaccines against CSF and PR. Then, the research team further verified the possibility of preparing a trivalent vaccine with rPRVTJ-delgE/gI/TK. They used the Fosmid library platform to construct recombinant virus rPRVTJ-del*gE*/*gI*/*TK*-E2-Cap to co-express CSFV E2 protein and porcine circovirus type 2 (PCV2) capsid proteins based on their use on the PRVTJ skeleton with *gE*/*gI*/*TK* gene deletion [60]. The recombinant virus was stable for 20 generations. However, only anti-PRV antibodies, but not anti-PCV2 or anti-CSFV antibodies, were induced by the recombinant virus in animal studies. The authors suggested that it may be that the insertion sites for E2 and Cap were inappropriate or that the antigen expressed does not stimulate an immune response to produce neutralizing antibodies. The authors are set to continue to optimize the design to develop a trivalent vaccine.

Some studies have shown that porcine reproductive and respiratory syndrome virus (PRRSV) could be used as a viral vaccine vector. The continuous passage from HP-PRRSV HuN4-F5 led to the attenuated strain HuN4-F112. An exciting study used a reverse genetic strategy to insert a neutral epitope of the E2 gene into the Nsp2 region of the backbone of the PRRSV HuN4-F112 strain, but the protein expression level was low and the immune effect was not satisfactory [61]. Subsequently, some scholars proposed to insert the E2 gene between ORF1b and ORF2 of the PRRSV HuN4-F112-attenuated strain (rPRRSV-E2), and at the same time, add endogenous signal peptide at the 5-terminal of the E2 gene and delete the transmembrane region at the 3-terminal, to better express the E2 protein [62]. The results showed that the rPRRSV-E2 recombinant virus could be stably transmitted at least 25 times in MARC-145 cells. A single intramuscular injection of the rPRRSV-E2 vaccine could protect piglets from the lethal challenge of highly pathogenic (HP)-PRRSV and CSFV. The immunization duration was up to five months [63], and the pre-existing maternal antibody (MDA) did not affect the immune effect of the rPRRSV-E2 vaccine [64].

Swine pox virus (SPV) had a powerful packaging capacity for recombinant DNA. It could encode a large number of recombinant proteins that could induce an immune response and could be used to develop recombinant vaccines [65,66]. Expression of CSFV E2 protein in swine pox virus (SPV) conferred immunogenicity to pigs. Lin et al., inserted the E2 glycoprotein gene into the porcine pox virus (SPV) genome by homologous recombination to form a recombinant rSPV-E2 virus [67]. The csfv-specific neutralizing antibody was detected at 7 dpi in the rSPV-E2 immunized group, whereas it was not detected until 14 dpi in the commercial c-strain vaccine-immunized group. The csfv-specific neutralizing antibody titers were significantly higher in the rSPV-E2-immunized group than in the commercial c-strain vaccine-immunized group at each time point after immunization. The results showed that rSPV-E2 could induce humoral and cellular immune responses and protect pigs from viremia. It is a promising candidate vaccine.

Recombinant Newcastle disease virus (rNDV) has been proved to be a vector for expressing foreign genes of animal viruses [68,69,70]. Kumar et al., used recombinant Newcastle disease virus (rNDV) to express E2 and Erns proteins of classical swine fever in cell cultures and chicken embryos [71]. Both rNDV-E2 and rNDV-Erns vaccines induced neutralizing antibodies in pigs by intranasal vaccination, and the neutralizing antibody potency of rNDV-E2 was higher than that of rNDV-Erns. In addition, ELISA based on E2 and Erns proteins expressed by rNDV could be used to screen CSFV infection and distinguish pigs infected with CSFV from vaccinated pigs.

In another study, the potential of the C strain as a viral vaccine vector was evaluated. Zhang et al., constructed three recombinant viruses based on the C strain as viral vectors [72], one containing the capsid (Cap) gene of porcine circovirus type 2 (PCV2) with a nuclear localization signal peptide (NLS) (rHCLV-2ACap), and two others lacking NLS, one containing a lactococcal ubiquitin-specific peptidase gene (usp) signal peptide (pHCLV-uspCap) and the other containing a bovine prolactin gene (psp) signal peptide (pHCLV-pspCap). Antibodies against CSFV and PCV were detected in rabbits inoculated with rHCLV-uspCap and rHCLV-pspCap. In contrast, only CSFV antibodies were detected in rabbits immunized with rHCLV-2ACap, and no anti-Cap antibodies were present. This may be because NLS signaling impedes the expression of the cap protein, which the authors did not explain in depth.

### 3.3. Subunit Vaccine

Live swine fever vaccine C strain is a classic live vaccine, but with the increasing complexity of the breeding environment in China, its defects and deficiencies such as difficult quantification of antigen, serious interference of maternal antibody, immune tolerance, and inability to differentiate diagnosis have become increasingly apparent. Subunit genetic engineering of vaccines is a good solution to these problems, with high antibody level, easy identification, and good antibody uniformity, which is the only way to eradicate classical swine fever in the future. A subunit vaccine refers to a new type of vaccine in which the nucleic acid sequence of the conserved antigen of the pathogenic microorganism is cloned into bacteria or cells by genetic engineering. The antigen protein is efficiently expressed and used in combination with vaccine adjuvants. E2, the structural protein of CSFV, has good antigenic characteristics and is related to inducing the body to produce neutralizing antibodies [73], which has been widely used to develop subunit vaccines, and the subunit vaccines prepared from it can realize DIVA according to ELISA detection kits for Erns or NS3 antibodies [74]. It is reported that a series of E2-based subunit vaccines are produced by different expression systems. In the following, we will discuss the various expression systems that have been commonly used in recent years.

#### 3.3.1. Expression of the E2 Protein in the Insect Expression System

The baculovirus–insect expression system can correctly modify foreign proteins, such as glycosylation, phosphorylation, and disulfide bond formation [75,76], so that the expressed proteins can be immunogenic to animals. Insect cells grow in suspension, which is easy to scale up and culture and is conducive to the large-scale expression of recombinant protein. The E2 subunit-marked vaccine, Tian Wen Jing (TWJ-E2^®^), developed by Tecon Biology Joint Stock Company Ltd. (TECON, Shenzhen, China), based on a baculovirus expression vector system, is the first marker vaccine in China and was officially launched in 2018. The vaccine was produced in High Five insect cells by using the baculovirus expression vector system with E2 glycoprotein modified by the subtype 1.1 vaccine C strain as the immunogen. The DIVA principle of the vaccine was to prevent disease by detecting antibodies against E^rns^ protein to recognize immunized versus infected animals. The TWJ-E2 vaccine provided complete protection against the highly virulent genotype 1.1 Shimen strain after two doses [77]. In contrast, the currently prevalent strain of CSFV in China was genotype 2.1, which differs significantly in antigenicity from the commercial TWJ-E2 vaccine. To address this situation, Gong et al., immunized four-week-old piglets with the TWJ-E2 vaccine, followed by attack experiments with the genotype 2 (2.1 b, 2.1 c, 2.1 h, and 2.2) strains [78]. Pigs vaccinated with TWJ-E2 showed no clinical signs and produced higher levels of E2-neutralizing antibodies than those in the commercial C-strain vaccination group. The results showed that the TWJ-E2 vaccine provided complete immune protection against the genotypic heterozygous of CSFV.

#### 3.3.2. Expression of E2 Protein by the Plant Expression System

Plants have been used as the host for recombinant vaccine production for more than 25 years [79,80]. As early as 2006, a Newcastle disease (ND) vaccine prepared by plants was approved for market in the United States [81]. Plant expression systems have advantages such as low production costs, easy scale-up of production, transient nature of expression systems, and faster development time [82].

Recently, several researchers have focused on the systematic expression of antigens in plants and investigated their immunogenicity safety. An interesting study showed how plant expression systems can efficiently express E2 proteins [83]. They used a highly efficient 5′ UTR (untranslated region) sequence immediately upstream of the AUG in the chimeric gene to improve translation efficiency; the strong, double-enhanced CaMV S35 promoter was used to enhance translation efficiency, and the endoplasmic reticulum (ER) retention signal HDEL was added to induce protein accumulation in the ER, using the cellulose-binding domain (CBD) as a purification tag [83]. The final production of the E2 fusion protein in transgenic Arabidopsis plants was up to 0.7% of the total soluble protein, and the protein purity was 90%. Anti-CSFV E2 antibodies were detected in the sera of immunized mice. Subsequently, Park et al., reported that E2 protein (ppE2) was transiently expressed in the transgenic Nicotiana benthamiana plant [84]. Injections of 50 or 100 µg stimulated the production of anti-E2 antibodies. These data proved that CBD was highly immunogenic. Fusion expression of the porcine immunoglobulin IgG Fc structural domain with the antigenic protein enhanced the immune response of the target animal. Another study substituted the recombinant protein Fc structural domain of porcine IgG for CBD with E2 protein fusion expression (pmE2:pFc2), using Nicotiana benthamiana plant to express the protein [85]. Confirmation that the pmE2:pFc2 fusion exists as a multimer rather than a dimer and that the Fc structural domain enhances the solubility and expression of E2. A total of 302 mg of recombinant pmE2 protein was produced from 1 kg of tobacco leaf. This was approximately ten-fold higher than previously reported using CBD:pmE2 (30.3 mg/kg) [84]. Immunization of mice or piglets resulted in the production of anti-pmE2 antibodies. Laughlin et al., further investigated this. Based on an Agrobacterium-mediated transient expression platform for Benthamiana expressing E2, a novel oil-in-water emulsion adjuvant, KNB adjuvant, was used to formulate the purified antigen [86]. Robust anti-E2 IgG and CSFV-neutralizing antibody response were generated in vaccinated pigs, with reduced fever and viremia in the single-dose vaccination group. In addition, Xu et al., prepared an oral immunization vaccine using Lactobacillus Plantarum expressing the fusion protein L. plantarum/pYG-E2-Tα1, and immunized animals could produce IgG with high virus-neutralizing activity against classical swine fever E2, significantly enhancing cellular immunity [87]. This was because Lactobacillus could deliver antigens to the mucosal immune system [88,89], and thymosin α1 (Tα1) promotes lymphocyte maturation and enhances T-cell function [90]. Additionally, it could be used as an immunologic adjuvant. 

#### 3.3.3. Self-Assembled Nano-Vaccine

Nano-vaccine is a new type of vaccine that combines an antigen with NPs and displays it on its surface through chemical coupling or gene fusion. In the 1950s, researchers found a rod-shaped particle that did not contain genetic material from the tobacco mosaic virus, the earliest self-assembled protein case [91]. In the 1970s, the surface antigen HBsAg of hepatitis B virus was found: granular, non-nucleic acid, and non-infectious [92]. In 1986, the first hepatitis B vaccine containing recombinant nanoparticles (HBsAg) was licensed [93,94]. Until now, HBsAg is still used as a nanoparticle scaffold for clinical research. The method of preparing vaccines based on nanoparticles has gained people’s attention. In recent years, nanoparticles have attracted more and more attention in medicine because of their unique physical and chemical properties. The advantages of nanoparticles are: (1) biocompatibility, non-cytotoxicity, non-immunogenicity, and no effect on cell function; (2) promote the maturation (APCs) and migration of DCS improve the phagocytosis and internalization of antigens in DCS (dendritic cells); (3) promote the activation of innate immunity and the secretion of immune-stimulating cytokines [95], thus promoting the initialization of the immune response [96]. At present, gold nanoparticles (AuNPs), ferritin nanoparticles, GEM nanoparticles, and mi3nps have been studied and coupled with the CSFV E2 protein to prepare nano-vaccines with good results. Next, these nano-vaccines will be explained in detail.

In a recent study, gold nanoparticles, AuNPs, were coupled with the E2 protein to form a stable complex, E2-AuNPs. Cell experiments showed that the AuNPs vector was non-toxic to antigen-presenting cells and accelerated the uptake of E2 protein by antigen-presenting cells. Compared with E2 or AuNPs, the E2-AuNPs group could induce humoral and cellular immunity better. This was the first time E2 protein was coupled with AuNPs [97], and the immunogenicity was analyzed. It laid the foundation for developing subunit vaccines using nanoparticles as antigen carriers. A large number of studies have shown that self-assembled ferritin nanoparticles could produce a large number of neutralizing antibodies. Ferritin nanoparticles are very suitable for carrying and exposing immunogens, self-assembled into nanoparticles, and have good thermal and chemical stability. Zhao and colleagues studied the feasibility of a self-assembled Fe protein Np platform for CSFV antigen display and vaccine development [98]. Based on the baculovirus expression system, E2 protein was ligated to the NP terminal of ferritin by a Gly-Ser-Gly (GSG) linker, and the E2 protein was successfully displayed on the surface of the ferritin nano-platform and emulsified with water or water adjuvant (the water or aqueous adjuvant Gel02 (Montanide, Seppic™, Castres, France)) to immunize rabbits. The results showed that the level of CSFV E2-specific antibodies in the pe2-Fe/Gel02 group was significantly higher than that in other groups, which could stimulate a strong neutralizing antibody level and induce humoral and cellular immunity at the same time. The signal peptide SP synthesized by Ze-HuiLiu and his team fused with truncated classical swine fever E2 protein to form a SP-E2 protein. The SP-E2 fusion protein was ligated to the N-terminal of self-assembled peptide nanoparticles’ mi3 by GGS flexible peptide. The results showed that the SP-E2-mi3 fusion protein could be self-assembled into NPs. A single dose of 10 mg of SP-E2-mi3 NPs could provide protection against CSFV [99].

In addition to the two above self-assembled nano-vaccines, a research team designed synthetic peptides to combine with nanoparticles to prepare nano-vaccines. For example, Hu et al., first combined PL23 peptide ligands with Gram-positive enhancer matrix (GEM) nanoparticles, then assembled GEM-PL-E2 particles with E2 proteins and selected peptides as the bridge between GEM and antigen proteins to construct a novel GEM-PL nanoparticles antigen display system based on affinity peptide ligands [100]. The prepared GEM-PL-E2 particles had a good immune function, and the experimental results showed that GEM nanoparticles could promote the growth of APCs and the absorption of antigens. GEM-PL-E2 particles could induce mice to produce high levels of neutralizing antibodies and anti-classical swine fever antibodies. GEM nanoparticles could activate innate immunity and stimulate the maturation of dendritic cells [101], which was consistent with the results of others. Compared with the traditional GEM-PA system, the GEM-PL system constructed by the author had the advantages of being easy to use and easily obtained peptide ligands. Most importantly, there was no need to express PA (Proteinanchor protein Anchor) fusion protein alone, which was the most challenging step. In addition, the author used computer virtual screening technology to simulate the binding of virtual peptide ligands to proteins, predict the binding mode, and select the optimal peptide ligand [102]. The method is worthy of reference.

#### 3.3.4. Adjuvant

An essential step for subunit vaccines to achieve better immunogenicity is to choose an appropriate adjuvant, which can enhance the immune response of subunit vaccines, activate the immune regulation pathway, and improve the vaccine’s efficacy [103]. Therefore, selecting an appropriate adjuvant is the key to obtaining a high-efficiency CSF subunit vaccine. Traditional vaccine adjuvants can “wrap” antigens well and constantly stimulate the body to produce antibodies. However, the development of many new molecular adjuvants makes the subunit vaccine obtain twice the result with half the effort. New cytokines can induce the immune response of the body, activate specific cell subsets, and cause a broader range of cytokine responses, and better activate the immune response of targeted cells.

Numerous studies have shown that the co-expression or fusion expression of cytokines with protective antigens is often synergistic, and the induction of immunity is often superior to that of the antigen alone [104]. The CD154 molecule is a glycoprotein of the tumor necrosis factor family, and the reaction of CD154 with its receptor CD40 enhances the humoral and cellular immunity of the body. For this property of CD154, Suárez et al., used the E2 protein of the Margarita CSFV strain fused with the extracellular domain of CD154 to form the E2-CD154 chimeric protein. They produced the protein in suspension culture using HEK293 cell line at concentrations up to 50 mg/L [105]. The E2-CD154 vaccine was combined with the Montanide TM ISA50 V2 adjuvant to immunize gestating sows, and the results showed complete protection against classical swine fever virus at 7 dpv. The E2-CD154 subunit vaccine was safe and protects pigs against CSF. However, its safety and immunogenicity in pregnant sows and the ability of maternally sourced neutralizing antibodies (MDNA) to protect offspring have not been demonstrated. Perez et al., evaluated the safety and immunogenicity of E2-CD154 in gestating sows and the ability of MDNA to protect progeny [106]. To introduce the E2-CD154 vaccine into areas where MLV vaccination is available, a further study was conducted in which gestating sows were vaccinated with the candidate vaccine and high maternally derived antibody (MDA) titers were present in piglets born to sows vaccinated with E2-CD154. These MDA titers were kept above 1:200 for the first seven weeks of life. One week after the immunization of the piglets, protective neutralizing antibodies (NAb) were induced with potencies above 1:600, and protective NAb levels persisted until six months of age without adverse effects [107]. 

IFN-γ is the only member of the type II interferon family, which is mainly produced by activated T cells, and activated by NK cells and macrophages [108]. Several studies have shown that IFN-γ could be used as a new vaccine adjuvant. The co-expression of IFN-γ and antigen genes usually synergistically affected the antigen immune response [104,108,109,110]. The study was based on the baculovirus system that co-expressed the E2 protein of CSFV with IFN-γ. The study results showed no significant enhancement of both specific and neutralizing antibody titers to classical swine fever virus by IFN-γ compared to immunization with the E2 protein alone. However, co-administration of E2 and IFN-γ subunit vaccines significantly enhanced the expression of classical swine fever virus-specific IFN-γ [111].

Research has demonstrated that the KNB adjuvant could enhance the immune effect of the CSFV subunit vaccine [86,112,113]. Madera et al., prepared subunit 54 containing the E2 protein expressed by insect cells and KNB oil-in-water emulsion, which could protect against CSF symptoms in a single dose containing 75 μg recombinant E2 protein. Subsequently, Madera further studied KNB-E2, and animal experiments showed that a single dose of KNB-E2 containing 25 μg of recombinant E2 protein could prevent CSF [113]. In this experiment, the CSFV of genotype 1.1 was studied, but most of the prevalent strains in the world are genotype 2. The authors will do follow-up studies to determine the efficacy of KNB-E2 on genotype 2.

Then, Burakovab et al., reported that another subunit vaccine was prepared by mixing the novel OWq saponin-based emulsion with E2 protein expressed by insect cells. OWq adjuvant, which is a mineral oil-based emulsion with a food-grade inexpensive saponin extract (OWq), and an oil-based adjuvant that produces an emulsion after gentle mixing with an aqueous phase were tested for safety and immunological activity in swine vaccination [114]. In order to further verify the efficacy of OWq, the authors carried out a series of animal experiments. The experimental results showed that the two-dose vaccination with CSFV glycoprotein E2-based vaccine formulated with OWq produced higher levels of E2-specific IgG and virus-neutralizing antibodies in pigs in contrast with animals that received the vaccine adjuvanted with oil only. These data indicated that the new OWq adjuvant was safe for pig immunization.

It is of great significance to establish the insect cell baculovirus expression system of the E2 protein and improve the secretion and expression level of the E2 protein for related research. Signal peptides could promote protein secretion to the outside of the cell. Researchers have achieved the secretory expression of the E2 protein by introducing signal peptides. The melittin signal peptide [115], the gp67 signal peptide, and the immunoglobulin kappa (Igκ) signal peptide [116] have been studied for secretory expression of E2. Xu et al., expressed a novel signal peptide SPZJ (SP23) fused to a novel CSFV E2 sequence (E2ZJ) isolated from an endemic strain from Zhejiang, and the level of E2 protein secretion induced was at least 50% greater than that caused by other signal peptides [117]. A single injection of 5 μg of E2ZJ could induce protective antibodies in piglets, which had 100% protective effect against lethal virus attack. The experimental results showed that SPZJ-E2ZJ was a promising candidate vaccine for the classical swine fever virus subunit vaccine.

## 4. Conclusions

In recent years, the epidemic characteristics, clinical symptoms, and pathological changes of CSF have gradually changed, characterized by low morbidity and mortality, recessive infection, mixed infection, and sporadic regional epidemic. This change seriously restricts the development of the Chinese pig industry and brings significant challenges to the prevention, control, and eradication of CSF. For a long time, large-scale vaccination of a live-attenuated vaccine has effectively controlled the spread of CSF. Still, due to the lack of suitable serum markers of the live-attenuated vaccine, it is difficult to distinguish between infected animals and vaccinated animals, which creates difficulties for the eradication of CSF. Therefore, the new vaccine for CSF should have the safety, efficiency, and low cost of the attenuated vaccine for CSF based on the C strain and have the function of distinguishing infected animals from vaccinated animals. In recent years, various scientific research institutions have been trying to explore the possibility of a new CSF marker vaccine and have made good progress (see Table 1). The successful implementation of marker vaccines is inseparable from accurate differential diagnosis. There are existing detection methods for CSF marker vaccines on the market, but they are insufficient in sensitivity and specificity and need to be further improved. The diagnostic methods of DIVA matched with vaccines have great potential for development. The safe and efficient DIVA vaccines combined with accurate diagnostic techniques and biosafety measures are expected to better control and eradicate CSF. 

## Figures and Tables

**Figure 1 vaccines-10-00603-f001:**
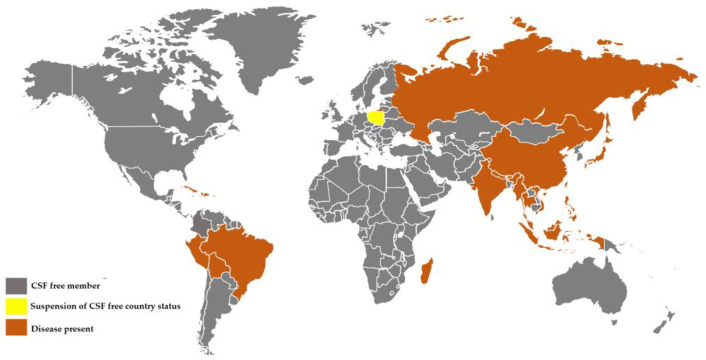
Global distribution of CSF epidemics in 2021. Romania was delisted from its CSFV-free country status in 2020. Map based on data from the CABI Invasive Species Compendium and official data from the OIE (accessed on 5 March 2022).

**Table 1 vaccines-10-00603-t001:** A review of marker vaccines against CSFV published in the last few years.

Type of Vaccine	Vaccine	Marker	Results and References
Live marker vaccine	CP7_E2Alf	E^rns^	Resistance to wild virus infection of genotypes 2.1 and 2.3 [29].
Live marker vaccine	Ra, Pro, RaPro	E^rns^	Protection of animals from CSFV infection 28 days after a single vaccination. No cross-reactivity in serological diagnosis [34].
Live marker vaccine	FLC-LOM-BE^rns^	E^rns^	Complete protection for gestating sows and increased productivity [35,36].
Live marker vaccine	rHCLV-E2P122A	^116^LFDGTNP^122^ epitope, recognized by the mAb HQ06	Intramuscular injection induces neutralizing antibody production at 28 days [37].
Live marker vaccine	FlagT4Gv	Flag epitope or mAbWH303 epitope	Protective effect on day 3 after inoculation and increased IFN-α levels in immunized animals [39,40]
Viral vector vaccine	rAdV-SFV-E2	E^rns^	Two doses of 6.25 × 10^5^ TCID50 or single dose of 10^7^ TCID50 provided complete protection against the challenge of deadly CSFV, and maternal antibodies did not inhibit the efficacy of the vaccine [51,52].
Viral vector vaccine	rPRVTJ-del*gE*/*gI*/*TK*-E2	E^rns^	Induced production of anti-CSFV and anti-PRV neutralizing antibodies, and complete protection against CSFV Shimon strain and variant PRV TJ strain attacks [59].
Viral vector vaccine	rPRRSV-E2	E^rns^	A single intramuscular injection protects piglets from the lethal challenge of highly pathogenic (HP)-PRRSV and CSFV, and the immunity lasts for up to 5 months [63,64].
Viral vector vaccine	rSPV-E2	E^rns^	Immunization at 7 dpi can detect csfv specific neutralizing antibody and induce humoral and cellular immune responses [67].
Viral vector vaccine	rNDV-E2	E^rns^	Intranasal inoculation induces the production of neutralizing antibodies against CSFV [71].
Subunit vaccine	TWJ-E2^®^	E^rns^	Two vaccinations provide complete protection against the highly virulent genotype 1.1 Shimen strain and protection against genotype 2 [77,78].
Subunit vaccine	ppE2-CBD	CBD	CBD-E2 fusion protein had high immunogenicity to piglets. A total of 50 or 100 µg injection could produce anti-E2 antibodies [84].
Subunit vaccine	pmE2:pFc2	E^rns^	Production of 302 mg of recombinant pmE2 protein in 1 kg of tobacco leaves. A single dose of l µg of vaccine is sufficient to induce immune responses in mice [85].
Subunit vaccine	L. plantarum/pYG-E2 -Tα1	E^rns^	Oral immunization produces anti-CSFV E2 IgG with high viral neutralizing activity, which significantly enhances cellular immunity [87].
Subunit vaccine	E2-AuNPs	E^rns^	The AuNPs vector is non-toxic to antigen-presenting cells, and the combination of E2 and AuNPs provides better induction of humoral and cellular immunity [97].
Subunit vaccine	pE2-fe/Gel02	E^rns^	Stimulates strong levels of neutralizing antibodies and can induce both humoral and cellular immunity [98].
Subunit vaccine	SP-E2-mi3 NPs	E^rns^	A single dose of 10 µg of SP-E2-mi3 NPs provides clinical protection against a CSFV challenge. Cross-protective for different genotypes [99].
Subunit vaccine	GEM-PL-E2	E^rns^	GEM-PL-E2 particles promote innate immune responses and induce higher neutralizing antibodies and anti-CSFV antibodies than CSFV E2 protein [100].
Subunit vaccine	E2-CD154	E^rns^	Complete protection against classical swine fever virus at 7 dpi, preventing vertical transmission, and the CD154 molecule enhances cellular immunity [105,106,107]
Subunit vaccine	E2-IFN-γ	E^rns^	Combination of E2 and IFN-γ significantly enhances expression of classical swine fever virus-specific IFN-γ [111].
Subunit vaccine	SPZJ-E2ZJ	E^rns^	The level of E2 protein secretion induced was at least 50% higher than that induced by other signal peptides, and a single injection of 5 μg of E2ZJ induced protective antibodies in piglets [117].
Subunit vaccine	KNB-E2	E^rns^	A single dose of KNB-E2 containing 25 µg of recombinant CSFV E2 protein can prevent CSFV genotype 1.1 in pigs. Produces higher levels of E2-specific IgG and virus-neutralizing antibodies [113].
Subunit vaccine	OWq-E2	E^rns^	Two doses of vaccination produced high levels of E2-specific IgG and virus-neutralizing antibodies in pigs [114].

## Data Availability

No new data were created or analyzed in this study. Data sharing is not applicable to this article.

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
