# Peer review of "The Development of Classical Swine Fever Marker Vaccines in Recent Years"

_vaccines, 2022, doi:10.3390/vaccines10040603_

Round 1
Reviewer 1 Report
The authors well emphasise that it is always extremely important for the scientific community to have tools that distinguish between a recrudescence of a disease that has already been eradicated and vaccine positivity. For a long time, large-scale vaccination with live attenuated vaccine effectively controlled the spread of CSF. The authors also describe well the consequences of the lack of adequate serum markers of the live attenuated vaccine, so that it is difficult to distinguish between infected and vaccinated animals. In addition, the authors in the flow chart comprehensively list the possible molecules for the new CSF vaccine and the safety, efficiency and low cost of the attenuated classical swine fever vaccine based on the C strain, which when conjugated with the DIVA system, would make it possible to distinguish infected from vaccinated animals.
Author Response
Dear editors
Those comments are all valuable and very helpful for revising and improving our paper, as well as the important guiding significance to our researches. We have studied comments carefully and have made correction which we hope meet with approval.We appreciate for your warm work earnestly, and hope that the correction will meet with approval.
Line 19: “Classical swine fever (CSF)” …. Add the acronym.
Line 22replace “where the disease has been decontaminated” with “”where it had been eradicated”
Line 24: use “fight” instead of “decontaminate”
Line27: use “eradication” instead of description
Line 56: replace “and” at the beginning of the line with a comma.
Line 38: remove “then”
Line 43: replace “was” with “is”
Line 56: replace “distributed” with “found”
Line 58: use “bonification” instead of “purification”
Line 58: replace “Using the method of purifying CSF is….” With “Bonification is…”
Line 63: use“…to the prevention and the control of the disease”instead of “the prevention and control”
Line 65: replace “coexist” with “cause”
Line 70: “RIG-I-like receptor (RLR) signaling”
Line 79: “….better diagnosis, prevention and control of ASF”,add the comma
Line 83: remove the word “including”
Line 86: replace “provide a methodology” with “suggest a possible methodology”
Line 124: remove “and stimulate the body to produce specific immunity, thus protecting the organism.
Line 126: add a comma after “Currently”
Line 129:remove “muscular”
Line 135:add “toll-like receptors”
Line 143: use“there were” instead of “there will be”
Line 145: use plural….”vaccines”. “ Single vaccines, combined vaccines and triple vaccines”
Line 156: “eradication” instead of “purification”
Line 192: replace “exciting” with “extensive”
Line 236: replace “of” with “for”
Line 238: add the word “system” after the word “expression”
Line 247: use “etc.”instead of “and so on
Line 247:use “most importantly, it does not integrate into the host genome”instead of “the most important is safe will not integrate into the host gene”
Line 252: remove the dash line between “vaccine” and “induced”
Line 257: replace “They”with “they”
Line 267: replace “was” with “is”
Line 268: add“pseudorabies virus”before the word “ (PRV)”
Line 270: add “pseudorabies”before the word “(PR)”
Line 292: add “porcine reproductive and respiratory syndrome virus”before the word “PRRSV”
Line 294:replace “obtained” with “led to”
Line 299: replace “protein” with “gene”
Line 336: Delete the last letter of the world “could”
Line 338: replace “had immunogenicity” with “conferred immunogenicity”
Line 346: replace “was” with “is”
Line 365: replace “didn’t” with “did not”
Line 368: replace “the defects..” with “its defects….”
Line 372: replace “purify” with “eradicate”
Line 381: “In the following, we will discuss the various expression systems that….”
Line 392: replace “insect baculovirus” with “baculovirus-insect”
Line394: replace “are immunogenic” with “can be immunogenic”
Line 398: “marker vaccine”
Line 442: replace “were” with “was”
Line 466:use “immunologic adjuvant”instead of “immune adjuvant”
Line 510: replace“synthetic”with“csynthetic”
Line 529: add “s”
Line 565: “affected”
Line 574: remove “have proved”
Line 582: use“The authors will do follow-up studies to determine the efficacy of….” instead of “The follow-up authors will continue to study the efficacy….”
Line 589: use “the authors”instead of “we”
Line 605: “showed”
Line 612: use “eradication” instead of “purification”
Line 615: use “eradication” instead of “purification”
Line 620-623: use “The successful implementation of marker vaccines is inseparable from accurate differential diagnosis. There are existing detection methods for CSF marker vaccines on the market, but they are insufficient in sensitivity and specificity and need to be further improved”instead of “However…..”
Reviewer 2 Report
This review by Fangfang Li et al. provides an account of all the classical swine fever vaccines developed in recent years.
The review is quite exhaustive and detailed. As such, it provides the reader with a good overview of the topic. However, while this review is sufficiently detailed to inform the reader of the different CSF vaccine development strategies adopted to date, I think that it would greatly benefit from an extensive English language revision.
In the following, I have added some of the language issues that I believe should be changed (not necessarily following my suggestions though….).
Line 18: “Classical swine fever (CSF)” …. Add the acronym.
Line 22: “decontaminated” is a wrong word in this context. Do you mean, “eradicated”? If so, please replace “where the disease has been decontaminated” with “”where it had been eradicated”
Line 24: same as before, use “eradicate” instead of “decontaminate”. Or maybe use “fight” here, so that you do not repeat “eradicate” too many times.
Line26: “eradication”
Line 35: replace “and” at the beginning of the line with a comma.
Line 38: please remove “then”
Line 42: please replace “was” with “is”
Line 45: replace “distributed” with “found”
Line 47: use “bonification” instead of “purification”
Line 47: please replace “Using the method of purifying CSF is….” With “Bonification is…”
Line 53: “…to the prevention and the control of the disease”
Line 54: replace “coexist” with “cause”
Line 59: “RIG-I-like receptor (RLR) signaling”…..i.e. define the acronym
Line 68: “….better diagnosis, prevention and control of ASF”
Line 73: remove the word “including”
Line 75: please replace “provide a methodology” with “suggest a possible methodology”
Line 83: please remove “and stimulate the body to produce specific immunity, thus protecting the organism. Since you are talking about a vaccine, this phrase is redundant.
Line 85: add a comma after “Currently”
Line 89: “muscular strains” is most likely a mistake. You probably mean just “strains”
Line 102: instead of “there will be”, you probably mean “there were”
Line 103-104: use plural….”vaccines”. “ Single vaccines, combined vaccines and triple vaccines”
Line 115: “eradication” instead of “purification”
Line 140: replace “exciting” with “extensive”
Line 186: replace “of” with “for”
Line 187: add the word “system” after the word “expression”
Line 189: please do not use expressions such as “and so on”. This is too generic and colloquial.
Line 189: “most importantly, it does not integrate into the host genome”
Line 195: remove the dash line between “vaccine” and “induced”
Line 199: “they”, without capital T.
Line 209: replace “was” with “is”
Line 210: “pseudorabies virus (PRV)”
Line 211: please define “PR”
Line 234: replace “obtained” with “led to”
Line 240: if you are talking about the E2 protein, you should not use DNA terminology. Rather, refer to N-terminus (5’ end in DNA terms) or C-terminus (3’ end in DNA terms).
Line 248: “could”, not “couldd”
Line 250: replace “had immunogenicity” with “conferred immunogenicity”
Line 250: replace “was” with “is”
Line 277: replace “didn’t” with “did not”
Line 280: replace “the defects..” with “its defects….”
Line 284: replace “purify” with “eradicate”
Line 293: “In the following, we will discuss the various expression systems that….”
Line 296: replace “insect baculovirus” with “baculovirus-insect”
Line 298: replace “are immunogenic” with “can be immunogenic”
Line 302: you mean, “marketed”?
Line 347: replace “were” with “was”., or remove the A at the beginning of the phrase (line 346)
Line 354: “immunologic adjuvant”
Line 399: “synthetic”, not “csynthetic”
Line 415: “is”, not “s”
Line 453: “affected”
Line 460: remove “have proved”
Line 468: “The authors will do follow-up studies to determine the efficacy of….”
Line 475: who is “we” here? Burakovab et al or the authors of this review? In the first case, please change “we” with “the authors”. In the second case, please add the reference to this follo-up study, or, if not published yet, write “(data not shown)”.
Line 491: “showed”
Line 499: use “eradication” instead of “purification”
Line 502: same as above
Line 507-508: “However…..”. This phrase is not complete.
Author Response

(The authors gave the same response as above.)

Round 2
Reviewer 2 Report
Although the manuscript has been significantly improved, I would strongly recommend a final round of editing by a native English speaker. This would give the manuscript a better chance of being fully appreciated by the readership of Vaccines and being cited by the scientific community.